# Data-driven contact structures: From homogeneous mixing to multilayer networks

**Alberto Aleta**[1]*, **Guilherme Ferraz de Arruda**[1], **Yamir Moreno**[1,2,3]

**1** ISI Foundation, Turin, Italy, **2** Institute for Biocomputation and Physics of Complex Systems (BIFI), University of Zaragoza, Zaragoza, Spain, **3** Department of Theoretical Physics, University of Zaragoza, Zaragoza, Spain

* albertoaleta@gmail.com

**Data Availability Statement:** All data used in this paper comes from the POLYMOD study, which was published with the title "Social Contacts and Mixing Patterns Relevant to the Spread of

## Abstract

The modeling of the spreading of communicable diseases has experienced significant advances in the last two decades or so. This has been possible due to the proliferation of data and the development of new methods to gather, mine and analyze it. A key role has also been played by the latest advances in new disciplines like network science. Nonetheless, current models still lack a faithful representation of all possible heterogeneities and features that can be extracted from data. Here, we bridge a current gap in the mathematical modeling of infectious diseases and develop a framework that allows to account simultaneously for both the connectivity of individuals and the age-structure of the population. We compare different scenarios, namely, i) the homogeneous mixing setting, ii) one in which only the social mixing is taken into account, iii) a setting that considers the connectivity of individuals alone, and finally, iv) a multilayer representation in which both the social mixing and the number of contacts are included in the model. We analytically show that the thresholds obtained for these four scenarios are different. In addition, we conduct extensive numerical simulations and conclude that heterogeneities in the contact network are important for a proper determination of the epidemic threshold, whereas the age-structure plays a bigger role beyond the onset of the outbreak. Altogether, when it comes to evaluate interventions such as vaccination, both sources of individual heterogeneity are important and should be concurrently considered. Our results also provide an indication of the errors incurred in situations in which one cannot access all needed information in terms of connectivity and age of the population.

## Author summary

Disease modeling has experienced a substantial advance in the last decades. However, state-of-art models still lack a full representation of all possible levels of heterogeneity. Here, we compare several frameworks that either use the connectivity, the demography, or both features. Specifically, we analyze four scenarios: (i) two homogeneous mixings, considering either social or demographic data and (ii) two network models, one accounting only for the connectivity distribution and another that includes both connectivity and

Infectious Diseases" and it is available at doi.org/10.1371/journal.pmed.0050074.

**Funding:** YM acknowledges partial support from the Government of Aragon, Spain, through grant E36-17R (FENOL), and by MINECO and FEDER funds (FIS2017-87519-P). AA, GFdA, and YM acknowledge support from Intesa Sanpaolo Innovation Center. The funders had no role in study design, data collection, and analysis, decision to publish, or preparation of the manuscript.

**Competing interests:** The authors have declared that no competing interests exist.

demography. Our analyses highlight the differences between each approach and the role of demographic and connectivity distributions; while the contact pattern is crucial for the determination of the epidemic threshold, the age-structure is fundamental after the outbreak. Notably, regarding vaccination, both types of heterogeneity play a significant role, suggesting that none of them should be neglected for this purpose. Finally, our results provide estimates of possible errors when data about sources of heterogeneity is not available.

## Introduction

One of the most fundamental concepts in epidemic dynamics is the heterogeneity in the ability of hosts to transmit the disease. This heterogeneity can be described as a function of three components: an individual's infectiousness, the rate at which she contacts susceptible individuals, and the duration of the infection [1]. Of these three components, the second one is probably the hardest to correctly estimate since it depends on several factors not related to the pathogen itself, such as the demographic structure of the population or its contact patterns. Hence, the heterogeneity in the mixing patterns between individuals is a key element for the correct assessment of the impact of epidemic outbreaks [2, 3].

The heterogeneity of the population can be characterized by different degrees of resolution [4]. The most basic approach, known as homogeneous mixing, considers that a contact between any two individuals in a population occurs randomly with equal probability [5]. In the decade of 1980, due to the interest in studying the spreading of sexually transmitted diseases, this assumption had to be modified [6]. The population was then divided into groups according to some characteristics, such as gender or sexual activity levels, and the interaction between those groups was encoded in a contact matrix [7]. Even though a homogeneous mixing component was still present inside each group, because all individuals within a group were indistinguishable, this approach demonstrated that a core group of 20% of the individuals in the host population could lead to 80% of the transmissions, which called for a complete redefinition of disease control programs [8].

The disproportionate role that highly active individuals had in the spreading dynamics was mathematically encoded in the fact that the transmission did not depend on the average number of new partners but on the mean-square divided by the mean [9], being one of the earliest signs of the crucial role that heterogeneities play in the spreading of diseases. This approach was also applied to other types of diseases in which groups of hosts could be easily identified, including vector-borne diseases [10] or age-dependent diseases [11]. The importance of heterogeneous mixing patterns was thus acknowledged, and several empirical studies measured them for sexually transmitted diseases [12, 13]. Yet, data on the mixing patterns of the population determinant for the spread of airborne infectious diseases, and in particular, their relationship with the age of individuals was not collected at large scale until 2008, i.e., 20 years later [14].

A further step to include more heterogeneities in the system is to consider the complete contact network of the population, which contains explicitly who can contact who [15, 16]. This approach is of particular importance for airborne infectious diseases since it is not possible to define a priori groups of highly infectious individuals, such as the core group for sexually transmitted diseases. Moreover, this approach gives a simple explanation to super-spreading events, which attracted a lot of attention after the 2003 SARS pandemic [17, 18] and are currently being scrutinized again in the context of the COVID-19 pandemic [19]. It was observed that it was common to find hosts who transmitted the disease to many more individuals than

the average. Within a network perspective, this is just a consequence of the higher number of contacts, or degree, that some individuals have in the network [17, 18, 20]. This individual heterogeneity also signaled that outbreaks could be really large if key individuals become infected and, at the same time, gave a new target for efficient control strategies such as vaccinating highly connected individuals [21, 22]. However, despite the many advantages of this approach, determining the complete contact network of a large population is almost infeasible, especially for infections transmitted by respiratory droplets or close contacts. Hence, it is common to use idealized networks built using some empirical data of the population, such as the degree distribution [23].

Lastly, there are high-resolution approaches that rely on lots of statistical data to build agent-based models in which the behavior of every single individual is taken into account [24–29]. Note, however, that in agent-based models, individuals are usually assigned to certain mixing groups (i.e., their household, school, or workplace), and that inside those groups homogeneous mixing is used, due to the lack of data for all these settings at a country scale [30]. An important step to create more realistic models in this direction is to collect high-resolution data on individual contacts using wearable sensors [21], that can be used to build time-varying networks in which not only the information about who contacts who is contained but also the duration and frequency of contacts [31]. Several settings have been monitored, such as schools and workplaces [32, 33], or even conferences and museums [34, 35]. Although the data is still too rare to be used in large scale simulations, it has already been shown that the heterogeneity induced by the time-varying networks inside each mixing group produces a different outcome than the one obtained assuming homogeneous mixing within each group [30].

Our goal in this paper is to analyze the role of one particular type of heterogeneity in disease dynamics, namely, the age structure of the population. Originally, age was introduced into the models to study childhood diseases [5]. The classical approach consists of dividing the population into different groups, one for each age bracket under consideration, and establishing an age-dependent transmission rate. This transmission rate can be arranged in a matrix in which each element encodes the transmission probability between groups $i$ and $j$ (this matrix is also known as the Who Acquired Infection from Whom matrix [36, 37]). It is also possible to separate the effect of the transmission itself in a common parameter and encode the number of contacts between each group in the matrix [38]. Note that this procedure falls into the second category described previously. That is, it takes into account the heterogeneity induced by having different classes of individuals but hides the individual variability under a homogeneous mixing approach within each group, as in models of sexually transmitted diseases with groups with different activity levels. Nevertheless, this approach is widely used today and has yielded outstanding results for many diseases such as chickenpox [39], herpes zoster [40], measles [41–43], pertussis [44] and tuberculosis [45]. In fact, even though the theoretical basis of this method is relatively old, data on the contact patterns of the general population as a function of their age have been available only recently.

The first large-scale study on the contact patterns between and within groups in the context of infections spread by respiratory droplets or close contact took place in 2008 and was focused in Europe [14]. Since then, a number of studies covering different countries have appeared, although data on Africa and Asia are still scarce [46]. Various methods have been developed to infer the contact patterns in the absence of direct data [47–49], and to project them into the future [50]. And yet, most studies that use this data disregard the whole distribution of contacts and use only the average number of contacts between groups, completely neglecting the individual heterogeneity (with few exceptions [51]). As a consequence, in these studies, superspreading events cannot occur naturally, unless the model is modified, contrary to network models in which the large connectivity of some individuals can result in the appearance of

such events. Similarly, the virtual absence of an epidemic threshold for certain types of contact networks cannot be observed with these simplified contact patterns [52]. To bridge this gap, in this paper, we focus on analyzing the role that disease-independent heterogeneity in host contact rates plays in the spreading of epidemics in large populations under several scenarios, both numerically and analytically. Furthermore, in contrast to previous approaches to this problem [53–56], we use a data-driven approach to highlight not only the role of those heterogeneities but also to explore the validity of the conclusions that one can derive when only limited information about the population is available.

## Results

### Modeling the contact patterns of the population

There are multiple ways of modeling the contact patterns of the population, depending on the availability of data and the characteristics of the disease. In this work, we consider that diseases have the same outcome on all individuals regardless of their condition and that individuals do not change their behavior as a consequence of the disease. This way, we can focus on the effect of adding different characteristics to the population contact patterns.

To be more specific, we use the information from the survey that was carried out in Italy for the POLYMOD project [14]. In this project, over 7,000 participants from eight European countries were asked to record the characteristics of their contacts with different individuals during one day, including age, sex, location, etc. Since that pioneering work, the number of countries where this type of study has been conducted has been increasing steadily, but data on Africa and Asia are still scarce. Besides, the resolution and amount of information vary from study to study [46]. As such, we build four different models of interaction, assuming that only partial information about the population is available, see Fig 1.

The simplest formulation is the homogeneous mixing approach (model H), suitable when very limited information about the population is available. In this model, all individuals are able to contact each other with equal probability. The number of such interactions, $\langle k \rangle$, can be extracted from contact surveys simply by calculating the average number of contacts per individual. Note, however, that this formulation is very simplistic since all individuals are completely equivalent. A slightly better approximation is to divide the population into age-groups, given the demographic structure of the population, Fig 1B, and establish a different number of contacts between and within them (model M), which is the common approach currently used in the epidemic literature to model age-mixing patterns. In this case, the necessary information includes knowing the age of both individuals participating in each contact, although this information can be easily summarized in an age-contact matrix, $M$, where each entry $M_{\alpha\beta}$ represents the average number of contacts from an individual in age group $\alpha$ to individuals in age group $\beta$. Note that in both models only the average number of contacts is used, in one case the average over the whole population and in the other over each age-group.

Another possibility is to use the whole contact distribution, Fig 1D, to build the contact network of the population. This formulation is commonly found in the network science literature since it highlights the role that the disproportionate number of contacts of some individuals have in the dynamics of the disease. A simple way of creating these networks is to represent each individual $i$ as a node and extract its degree (number of contacts) from the distribution. Then, the expected number of edges between nodes $i$ and $j$ is $\langle \mathcal{A}_{ij} \rangle = k_i k_j / \sum_l k_l$ (model C). To obtain this expression, we can consider that each node $i$ has $k_i$ stubs associated. Next, if these stubs are matched together randomly, the probability that each stub from node $i$ ends up at one of the $k_j$ stubs of node $j$ is $k_j$ over the total number of stubs, $\sum_l k_l$. This method is known as the configuration model.

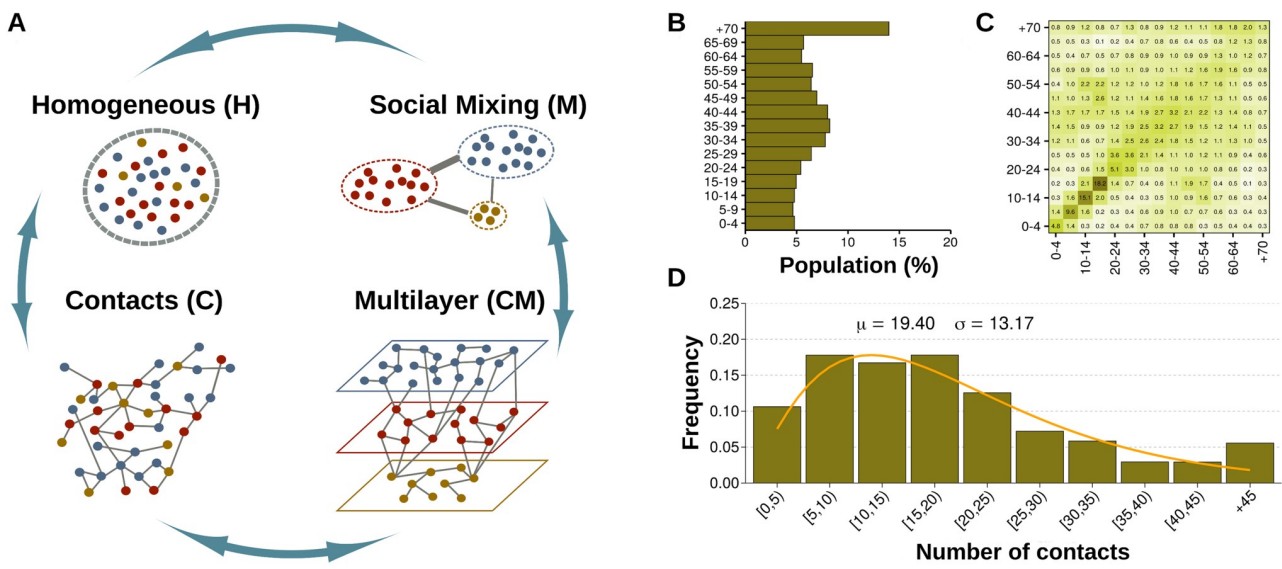

**Fig 1. Modeling the contact patterns of the population.** Panel A: Schematic view of the different models considered. If the only information available is the average number of contacts per individual, homogeneous mixing can be assumed (H). If there is information about the average number of contacts between individuals with age $a$ and $a'$, then a classical group-interaction model can be implemented (M). On the other hand, if the full contact distribution of the population is known, regardless of their age, it is possible to build the contact network of the population (C). Lastly, when both the contact distribution and the interaction patterns between different age groups are known, the individual heterogeneity and the global mixing patterns can be combined to create a multilayer network in which each layer represents a different age group (C+M). Panel B: Demographic structure of Italy in 2005 [57]. Panel C: Age-contact patterns in Italy obtained in the POLYMOD study [14]. Panel D: Contact distribution in Italy obtained in the POLYMOD study [14]. The $x$ axis represents the number of daily contacts and the $y$ axis the fraction of individuals that have reported such amount of contacts. The distribution is fitted to a right-censored negative binomial distribution since the maximum number of contacts that could be reported was 45.

Lastly, we can combine both ingredients, the mixing patterns, and the contact distribution of the population in a network representation. To do so, we propose to arrange nodes in a multilayer network, in which each layer represents an age-group. As such, the first step to create this network is to extract the age associated to each node from the demographic structure of the population, Fig 1B, and assign them to their corresponding layer (since we are working with 15 age-groups, our system is composed by that same amount of layers). Then, the degree of each node should be extracted from the desired distribution. To incorporate the mixing patterns into the configuration model, we propose the following scheme:

1. Given a node $i$ located in layer $\alpha$ (where the layer represents the age-group associated with $i$), the probability that any of its stubs ends up at a node in any layer $\beta$ (including the same layer) is $p_{\alpha\beta}$. This probability can be extracted from the mixing matrix as $p_{\alpha\beta} = M_{\alpha\beta}/\Sigma_\beta\, M_{\alpha\beta}$.

2. The stub from node $i$ will match the stub of node $j$, situated in layer $\beta$, with probability $k_j/\Sigma_{l\in\beta}\, k_l$, where the denominator indicates the addition over the degree of all nodes present in layer $\beta$.

Hence, the expected number of edges between nodes $i$ and $j$ will be given by

$$\langle \mathcal{A}_{ij} \rangle = k_i p_{\alpha(i),\beta(j)} \frac{k_j}{\sum_{l\in\beta(j)} k_l} \,. \tag{1}$$

Yet, note that incorporating the mixing patterns introduces a restriction in the degree distribution. Indeed, one of the important properties of the mixing patterns matrix is that it has

to verify reciprocity, i.e.,

$$M_{\alpha\beta}N_{\alpha} = M_{\beta\alpha}N_{\beta} \,. \tag{2}$$

That is, the number of contacts going from group $\alpha$ to group $\beta$ has to be the same as the ones from $\beta$ to $\alpha$ (if the populations of each group were equal, this would lead to a symmetric matrix). It is easy to see that Eq (1) only fulfills this property if

$$\sum_{l\in\alpha} k_l = \sum_{\beta} M_{\alpha\beta}N_{\alpha} \,. \tag{3}$$

And, thus,

$$\langle k \rangle_{\alpha} = \sum_{\beta} M_{\alpha\beta} \,, \tag{4}$$

where $\langle k \rangle_{\alpha}$ represents the average degree in layer $\alpha$. Hence, even though the shape of the distribution can be chosen freely, the mixing matrix fixes the average degree of each layer. Eqs (1) and (4) completely define our last model, the CM model, see Fig 1A.

## Susceptible-infected-susceptible dynamics

To determine the consequences of each of the previous assumptions, we first consider a general susceptible-infected-susceptible (SIS) Markovian model [58, 59]. In this model, the recovery rate of each infected individual is modeled by a Poisson process with rate $\delta$. In turn, each successful contact emanating from an infected individual (i.e., a contact that transmits the disease) is modeled as a Poisson process with rate $\lambda$. We denote by $Y_i$ the Bernoulli random variables that are equal to one if individual $i$ is infected or zero otherwise. Complementary, $Y_i + X_i = 1$ and $\sum_i X_i + Y_i = N$. The only ingredient left to be defined is how the contact process between individuals actually takes place. In general, in its exact formulation, we can do so by introducing the matrix $\mathcal{A}$, which denotes whether two individuals can contact each other or not [58, 59]:

$$\frac{d\langle Y_i \rangle}{dt} = \left\langle -\delta Y_i + \lambda X_i \sum_j \mathcal{A}_{ij} Y_j \right\rangle. \tag{5}$$

With this formulation we can already study the spreading of an epidemic on any network, models C and CM. Indeed, assuming that the states are independent, i.e., $\langle Y_i Y_j \rangle = \langle Y_i \rangle\langle Y_j \rangle \equiv y_i y_j$, we get

$$\frac{\mathrm{d}y_i}{\mathrm{d}t} = -\delta y_i + \lambda x_i \sum_j \mathcal{A}_{ij} y_j \,. \tag{6}$$

Considering that the nodes with the same degree are statistically equivalent, we can obtain the epidemic threshold using the heterogeneous mean field approximation [60],

$$\tau \equiv \frac{\lambda}{\delta} = \frac{\langle k \rangle}{\langle k^2 \rangle} \,. \tag{7}$$

This well-known result from network science clearly shows the importance of the heterogeneity of the contacts, since it depends on the second moment of the distribution. In the case of Italy, using this expression we obtain a theoretical threshold of $\tau_{CM} = 0.033$ and $\tau_C = 0.035$ for the CM and C models, respectively.

For the M model, since individuals are indistinguishable, Eq (6) is rewritten as

$$\frac{\mathrm{d}y_\alpha}{\mathrm{d}t} = -\delta y_\alpha + \lambda x_\alpha \sum_\beta M_{\alpha\beta} y_\beta \,, \tag{8}$$

where $M_{\alpha\beta}$ is the matrix depicted in Fig 1C, and $y_\alpha$ the fraction of infected individuals in layer $\alpha$. In this case, using the next generation approach [61, 62], the epidemic threshold is

$$\tau_\mathrm{M} \equiv \frac{\lambda}{\delta} = \frac{1}{\rho(M)} \,. \tag{9}$$

Regarding Italy, the spectral radius of $M$ is $\rho(M) = 22.51$, resulting in an epidemic threshold of $\tau_\mathrm{M} = 0.044$.

Lastly, the equation governing the H model is

$$\frac{\mathrm{d}y}{\mathrm{d}t} = -\delta y + \lambda \langle k \rangle xy \,, \tag{10}$$

where the epidemic threshold is

$$\tau_H \equiv \frac{\lambda}{\delta} = \frac{1}{\langle k \rangle} \,. \tag{11}$$

According to Fig 1D, the epidemic threshold in our system is thus $\tau_\mathrm{H} = 0.052$.

Thus, in this case, the following relation holds:

$$\begin{array}{ccccccc} \tau_\mathrm{H} & > & \tau_\mathrm{A} & > & \tau_\mathrm{C} & > & \tau_\mathrm{CM} \\ 0.052 & > & 0.044 & > & 0.035 & > & 0.033 \end{array} \tag{12}$$

Some observations are in order. First, even though the average number of contacts is the same in all models, the epidemic threshold is completely different. Besides, increasingly adding heterogeneity to the model lowers the epidemic threshold. This is especially relevant when going from classical mixing models to network models. Indeed, when we introduce the whole contact distribution, we are indirectly adding the possibility of having super-spreading events, which, as noted before, is missing in the classical approaches. On the other hand, as expected, the difference between both network models is relatively small ($\frac{\tau_\mathrm{CM}}{\tau_\mathrm{C}} = 1.06$) since the main driver of the epidemic threshold is the contact distribution. Nonetheless, as we shall see next, for other scenarios, the multilayer framework will yield quite different results from model C.

To asses the quality of our theoretical analysis, our first step is to obtain the epidemic threshold for each configuration numerically. To do so, we create an artificial population of $10^6$ individuals and assign them an age according to the demographic structure of the Italian population [57]. Then, we simulate a stochastic SIS Markov model, with $\delta = 1$ and multiple values of $\lambda$ for each of the four contact models under consideration (see Materials and methods). In Fig 2A, we show the attack rate (total number of cases over the whole population) as a function of $\lambda$. The overall behavior of the four scenarios is qualitatively similar, although large differences are observed in the value of the epidemic threshold (see inset), as predicted.

To properly characterize the value of the epidemic threshold and compare it with the theoretical expectations, we use the quasistationary state (QS) method [59, 63]. This technique allows computing the susceptibility of the system, which presents a peak at the epidemic threshold (see Materials and methods). The caveat is that it is highly dependent on the system size since the epidemic threshold is only properly defined for infinite systems. Nevertheless, in Fig 2B we compute the susceptibility, $\chi$ for the four configurations with system sizes ranging

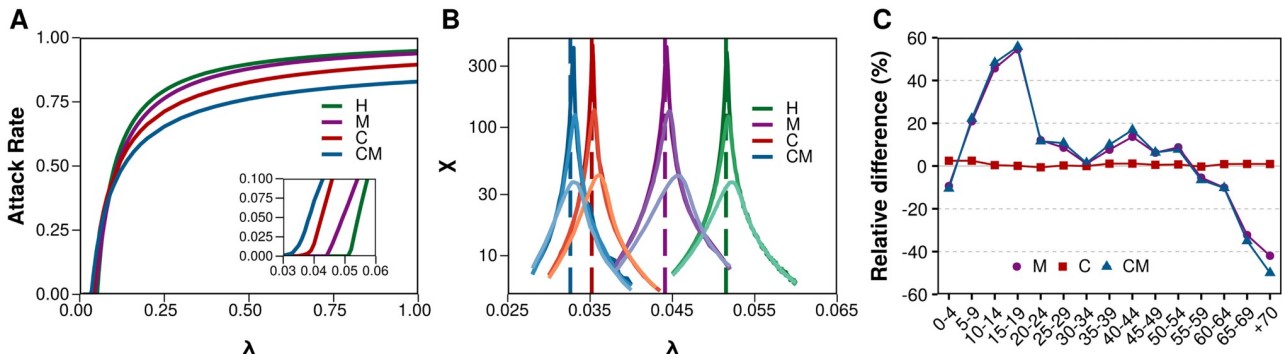

**Fig 2. Dynamics of a SIS model using different contact models.** A) The fraction of infected individuals as a function of the infection rate. In the inset, the area near the epidemic threshold for each configuration is shown enlarged. B) Susceptibility as a function of the infection rate for the four configurations with populations of size $10^4$, $10^5$ and $10^6$. The larger the size of the population the closer the peak of susceptibility is to the theoretical epidemic threshold (dashed line). C) Relative difference in the number of infected individuals between the results obtained using the M (purple circles), C (red squares) or CM (blue triangles) models and the homogeneous mixing setting. Positive values indicate that the number of infected individuals is larger than in the homogeneous mixing scenario, while negative values represent a lower number of infected individuals.

from $10^4$ to $10^6$ individuals and we can see that for the latter the peak of the susceptibility is already quite close to the predicted value of the epidemic threshold, validating our theoretical approach.

Next, we focus on studying the impact that the disease has on each age group under the different configurations, Fig 2C. We set the value of λ in each case so that the attack rate is equal to 0.4, since the four scenarios converge to that value for similar values of λ (see Fig 2A). Using the homogeneous mixing approximation, we obtain a distribution of infected individuals across ages proportional to the demographic structure of the population (Fig 1B), as one would expect given that all individuals are virtually indistinguishable for the dynamics. The same result is obtained for the C model, in which the age of the nodes is completely independent of the network structure. At variance with these results, if we incorporate the heterogeneous mixing patterns of the population either in the age-mixing (M) model or in the multilayer network (CM) setting, the incidence in each age group would be quite different, see Fig 2C. Note that we have again set λ so that the overall incidence is 0.40 in all cases –this assures that the total number of infected individuals is the same, only its distribution across age classes is different. Results show that in both scenarios the prevalence is much higher for teenagers and smaller for the older cohorts than in the homogeneous mixing model.

## Susceptible-infected-removed dynamics

Although the SIS model facilitates the theoretical and numerical analysis of the system, especially near the epidemic threshold, it is too simplistic to model real diseases such as ILI. Thus, to highlight the impact of these observations on a more realistic scenario, we slightly modify the model by incorporating the removed compartment so that the dynamics are governed by a susceptible-infected-removed (SIR) model, which is better suited for studying ILI [64].

It has been recently shown that using a constant and group-independent basic reproduction number, $R_0$, might not describe well key features of the disease dynamics in realistic scenarios [28]. For this reason, we first explore the dependency of this parameter with the age of the individual in the two networked scenarios. To do so, we simply count the total number of newly infected individuals that a single seeded infectious subject would produce in a fully susceptible population over $10^8$ simulations, with the value of λ set so that the average value of $R_0$ is 1.3 inline with typical values for influenza [65]. Fig 3A shows the value of $R_0$ as a function of the

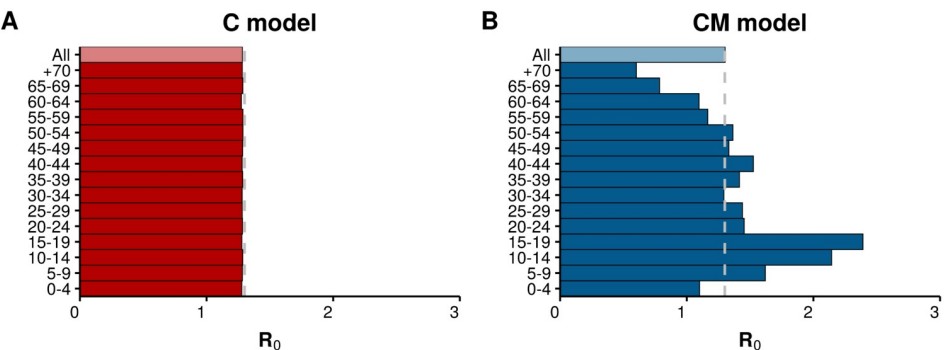

**Fig 3. Basic reproduction number in single layer and multilayer networks.** A) Measured value of $R_0$ in the C model, where both the degree distribution and the connections are completely independent of the age assigned to the nodes. B) The measured value of $R_0$ in the CM model, where the connection patterns follow the age mixing patterns of the population. In both cases the average $R_0$ of the total population has been set to 1.3.

age of the seed node in the network in which all nodes have the same degree distribution. Clearly, the same $R_0$ value is obtained regardless of the age of the nodes, as it should be given that both their degree and their connections are independent of their age. Conversely, in the multilayer network where the mixing patterns of the population are incorporated, Fig 3B, the situation changes completely. The value of $R_0$ is above the average for teenagers and adults but below the average for the elderly, highlighting the importance of the underlying structure in the value of $R_0$.

Lastly, we study the effect of vaccinating a fraction of the nodes before the epidemic begins. This sort of contention measures are among those that can benefit the most from knowledge about the structure of the population, as they allow devising more efficient vaccination strategies. First, we set the baseline scenario to values compatible with the 2018-2019 ILI epidemic in Italy. According to the World Health Organization, the total attack rate was 13.3%. Besides, an important fraction of the population was vaccinated preemptively. In Italy, vaccination is recommended for several groups of people, such as those with chronic medical conditions, firefighters, health care workers, or the elderly [66]. Of these groups, the only one that we can distinguish in our model is the elderly, but it is also the one with the largest vaccination rates. Unfortunately, the uptake of the vaccine has been decreasing for the past few years, and now is close to 50% [67]. Even more, the effectiveness of the vaccine is estimated to be around 60% yielding an effective vaccination rate of 30% in the elderly [68]. Hence, to obtain the baseline values in our model, we set 30% of the elderly in the recovered state initially and set the value of λ so that the attack rate is 13.3%, Fig 4A.

Our first observation is that in the C scheme, we trivially obtain a reduction in the attack rate among the elderly due to their vaccination, but otherwise, the incidence is the same in all age groups. On the other hand, both in the M and CM models, the attack rate depends highly on the age of the individual. To gauge the effect of increasing vaccination rates, we vaccinate 1% of the total population (assuming that the effectiveness is 60% for all age-groups). Note that since the elderly group represents 19% of the population, the initial vaccination rate was roughly 10% of the total population. If these new vaccines are administered randomly, we can see that the effect is just a homogeneous reduction of 5-6% in all age groups, independently of the model, Fig 4B.

Conversely, if that same amount of new vaccinations is targeted, the situation changes completely. In the M model, we vaccinate individuals belonging to the group with 15-19 years old since it is the one with the largest number of contacts and the highest attack rates. We can

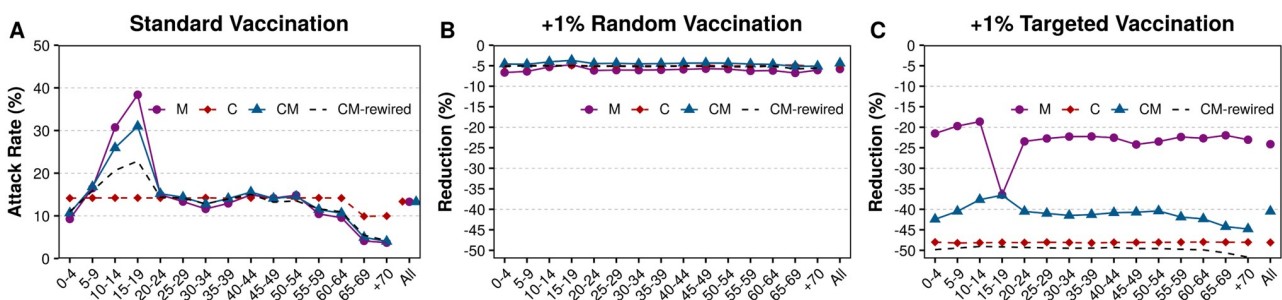

**Fig 4. Effects of different vaccination strategies.** A) Attack rate under the standard vaccination adjusted to the values of the 2018-2019 ILI epidemic. B) Reduction of the attack rate when the vaccination is increased by 1%*N* but applied randomly. C) Reduction of the attack rate when the fraction of vaccinated population is increased by 1%*N* but targeted to highly infective individuals: the group of 15-19 years old in the M model and the nodes with larger degree in the two networked scenarios.

see that the overall reduction is much larger than in the previous case, and especially so in this particular group, see Fig 4C. In the C and CM models, instead, we apply the vaccines to individuals with the largest degrees. We can see that the reduction is larger in the C setting than in the CM one. This result might seem counter-intuitive since the same measure is applied to both systems. However, note that while in the C model the largest degrees are homogeneously distributed across the population, in the CM model they are concentrated in specific age groups, or layers. Furthermore, since nodes in the same layer tend to be connected together, the previous observation implies that the effect of removing hubs will be lower. To verify this, we have rewired the connections of the CM model while preserving the age, degree and vaccination status of each node. As we can see, in such case we recover the same value as in the C model. In other terms, the correlations induced by the age mixing patterns lower the effectivity of this vaccination strategy. Note also that in both the random and the targeted vaccination schemes, the number of new vaccines introduced in the system is exactly the same, only who is vaccinated changes.

## Discussion

Models can range from simple homogeneous mixing models to high-resolution approaches. The latter, even though it might provide better insights, is also much more data demanding. As a compromise between the two, network models can capture the heterogeneity of the population while keeping the amount of data necessary low. Nevertheless, most network approaches focus only on determining the role that the difference in the number of contacts of the population has on the impact of disease dynamics but ignore other types of heterogeneities such as the age mixing patterns.

We have shown that to determine the epidemic threshold of the population properly, the heterogeneity in the number of contacts cannot be neglected, making the simple homogeneous approach and the homogeneous approach with age mixing patterns ill-suited for it. In fact, a description that ignores the age mixing patterns of the population can capture much better the value of the epidemic threshold. Furthermore, we observe two different regimes in the attack rate as a function of the spreading rate. For low values of the spreading rate, individual heterogeneity plays a more important role, yielding larger attack rates than the homogeneous counterparts. However, after a certain value, the phenomenology reverses, i.e., larger attack rates are obtained for the homogeneous approaches rather than for the networked versions. The reason is that, in homogeneous models, an infected agent can contact everyone in the population, and thus it can keep infecting individuals even if the attack rate is high. When the

network is taken into consideration, it is possible that nodes run out of susceptible individuals within their vicinity, virtually preventing them from spreading the disease any further.

On the other hand, if we study the distribution of infected individuals across age cohorts, we can see that the C scheme is no longer valid, yielding the same results as the simple homogeneous mixing approach. If the age mixing patterns are added into the model, either in the M or CM schemes, a larger fraction of young individuals will be infected, while the incidence in elder cohorts is reduced. Hence, even though the C approach can predict fairly well the value of the epidemic threshold, it cannot be used to study the spreading of diseases in which taking into account the age of the individuals is important beyond the epidemic threshold. Conversely, the multilayer network of the CM model can describe both the epidemic threshold and the distribution of the disease across age groups correctly. In other words, it combines both the importance that individual heterogeneity has with the inherent assortativity present in human interactions.

Individual heterogeneity also introduces important variations in the measured value of $R_0$. This observation is quite important since it shows that for the proper evaluation of $R_0$ during emerging diseases, the sampling of the population has to be done carefully. Biases in the sampled individuals, such as having too many young individuals, could lead to estimations of $R_0$ much larger than its actual value. Even more, this is not limited to the age of the individuals since we have also seen the importance of individual heterogeneity in the dynamics. Of utmost relevance, if in the sample, there are individuals with an average number of contacts higher than the normal population, the estimations of $R_0$ would also be higher.

Lastly, we have also observed the crucial role that heterogeneity plays if we want to devise efficient vaccination strategies. The role of networks in this regard is known to be important not only because there are tools that allow identifying the most important individuals, but because it provides a clear way to study herd immunity. Yet, if we do not take into account the contact distribution of the population the effectivity of vaccination campaigns will be lower. Conversely, if we rely simply on the contact distribution of the population and disregard their mixing patterns, we would overestimate the effect of vaccination.

As the current COVID-19 pandemic has shown, accounting for both the age and the contact heterogeneity of individuals is crucial to control the epidemic. It is yet unknown the exact role that age plays in this disease, although preliminary results show that children are less susceptible and that the case fatality rate for older individuals is much higher. Similarly, large super-spreading events are possible such as the ones detected in South Korea, Boston or Spain [19, 69]. The latter country is also among the ones most affected by the current epidemic, but empirical information about the age mixing patterns of the population is not available [46, 69]. Thus, to the inherent problems of forecasting the evolution of an emerging disease [70, 71] we have to add our ignorance about these factors which, as we have shown in this article, can substantially modify the predictions. This highlights once again the importance of obtaining precise information about the behavior of the population, enhancing our preparedness for this type of event.

To sum up, we have shown the importance that individual heterogeneities have on the spreading of infectious diseases. Yet, although in general the more details in the model the better, it is also important to take into account the inherent limitations about data that currently exist. Therefore, it is crucial to correctly gauge what can and cannot be done, given the information available to us. In particular, we have shown that to predict the epidemic threshold, it is indispensable to know the degree distribution of the population. Nonetheless, this is not strictly needed to evaluate the impact of a disease away from the threshold. Yet, adding this information, even though it does not dramatically change the predicted outcomes of the epidemic under normal conditions, could be pivotal to devise efficient vaccination strategies.

Furthermore, we have seen that the underlying information of the system also has an impact on quantities that are commonly measured and used in real settings, such as $R_0$, implying that care must be taken when extrapolating the results from one study to the other.

## Materials and methods

### Model

In all cases, we consider populations of $10^6$ individuals. In the H model, since individuals are indistinguishable, the impact of the disease over the age groups is computed by randomly extracting values from the demographic distribution of Italy in 2005 [57]. In the M model, the size of each age-group is computed using the same procedure. Besides, the age-mixing matrix was corrected so that reciprocity is fulfilled, and the average connectivity is exactly 19.40 [50]. In the C model, we randomly extract the degree of each node from a right-censored negative binomial distribution adjusted to the survey data from POLYMOD [14]. Then, links are sampled performing a Bernoulli trial over each pair of nodes respecting that $\langle \mathcal{A}_{ij} \rangle = k_i k_j / \sum_l k_l$. A similar procedure is followed to create the multilayer with age mixing patterns, but in this case, each layer has its own values for the negative binomial distribution, according to the data (see Fig 1D), and the probability of establishing a respects $\langle \mathcal{A}_{ij} \rangle = p_{\alpha(i),\beta(j)} k_i k_j / \sum_{l \in \alpha(j)} k_l$ where $p_{\alpha(i),\beta(j)}$ is the probability that a link from a node with the same age as node $i$ ends up at a node with the same age as node $j$, and $\alpha(j)$ is the layer to which $j$ belongs. We remark that the network is simplified, removing multiple edges.

### Epidemic threshold

Close to the critical point, the fluctuations of the system are often high, driving the system to the absorbing state [59, 63]. To avoid this problem, the quasistationary state (QS) method stores $M$ active configurations previously visited by the dynamics. At each step, with probability $p_r$, the current configuration (as long as is active) replaces one of the $M$ stored ones. Then, if the system tries to visit an absorbing state, the whole configuration is substituted by one of the stored ones. The system evolves for a relaxation time, $t_r$, and then the distribution of the number of infected individuals, $p_n$, is obtained during a sampling time $t_a$. Lastly, the threshold is estimated by locating the peak of the modified susceptibility $\chi = N(\langle \rho^2 \rangle - \langle \rho \rangle^2)/\langle \rho \rangle$, where $\langle \rho^k \rangle$ is the $k$-th moment of the the distribution of the number of infected individuals, $p_n$ (note that $\langle \rho^k \rangle = \Sigma_n n^k p_n$). In our analysis, the number of stored configurations and the probability of replacing one of them is fixed to $M = 100$ and $p_r = 0.01$, while the relaxation and sampling times vary in a range depending on the size of the system, $t_r = 10^4 - 10^6$ and $t_a = 10^5 - 10^7$.

## Author Contributions

**Conceptualization:** Alberto Aleta, Guilherme Ferraz de Arruda, Yamir Moreno.

**Data curation:** Alberto Aleta.

**Formal analysis:** Alberto Aleta, Guilherme Ferraz de Arruda, Yamir Moreno.

**Funding acquisition:** Yamir Moreno.

**Investigation:** Alberto Aleta, Guilherme Ferraz de Arruda, Yamir Moreno.

**Methodology:** Alberto Aleta.

**Project administration:** Yamir Moreno.

**Resources:** Alberto Aleta.

**Software:** Alberto Aleta.

**Supervision:** Guilherme Ferraz de Arruda, Yamir Moreno.

**Validation:** Alberto Aleta, Guilherme Ferraz de Arruda.

**Visualization:** Alberto Aleta.

**Writing – original draft:** Alberto Aleta.

**Writing – review & editing:** Guilherme Ferraz de Arruda, Yamir Moreno.

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
