## [Decision Letter · Decision Letter 0]

6 May 2020

Dear Dr. Aleta,

Thank you very much for submitting your manuscript "Data-driven contact structures: from homogeneous mixing to multilayer networks" for consideration at PLOS Computational Biology. As with all papers reviewed by the journal, your manuscript was reviewed by members of the editorial board and by several independent reviewers. The reviewers appreciated the attention to an important topic. Based on the reviews, we are likely to accept this manuscript for publication, providing that you modify the manuscript according to the review recommendations.

Sincerely,

Benjamin Althouse

Associate Editor

PLOS Computational Biology

Rob De Boer

Deputy Editor

PLOS Computational Biology

[LINK]

Reviewer's Responses to Questions

**Comments to the Authors:**

Reviewer #1: In "Data-driven contact structures: from homogeneous mixing to multilayer networks" authors bridge for the first time an important gap that has existed until now between data analysis and contact structures derived from this data, namely the simultaneous determination of both, the connectivity of individuals and the age-structure of the population. This advance has important implications for the mathematical modeling of infectious diseases, as beautifully shown in the paper. In particular, authors compare four different scenarios that gradually leads to taking into account a multilayer representation in which both the social mixing and the number of contacts

are included in the model. It is shown analytically that the thresholds obtained for these

four scenarios are different, and that indeed only the most comprehensive framework, as presented in the paper, allows for the correct determination of the epidemic threshold. This is further supported by systematic simulations, confirming further that heterogeneities in the contact network are vastly important and must not be overlooked if we wish for a proper

determination of the epidemic threshold. It is also shown that the age-structure, which is likewise determined by the new approach, plays a bigger role beyond the onset of the outbreak.

An accurate determination of epidemic thresholds in contact networks is of huge importance, both for mitigating and prediction epidemic spreading, as well as for devising effective vaccination strategies. This research points out clearly for the first time that, when it comes to the evaluation of interventions such as vaccination, both sources of individual heterogeneity are important and should be considered jointly. This was an important open problem in the realm of an intensely investigated subject with obvious practical ramifications. By introducing a clever new approach based on empirical data and network science, this study thus fills and incredibly important gap that bridges the divide, and it reveals just how wrong one could be by neglecting or not having access to all the needed information in terms of connectivity and age of the population.

The paper is well-written, comprehensive, and clear. I find it is among the finest papers that I have had the pleasure of reading in the recent past. The motivation behind the approach and the insights it affords towards improving spreading of communicable diseases is genius, and as such it will surely not fail to impress the diverse readership of PLOS Computational Biology. For these reasons, I warmly recommend publication.

It is quite a challenge to suggest improvements for such an excellent contribution. Perhaps a reference to the current COVID-19 pandemic and how the approach could improve forecasting, as studied in "Forecasting COVID-19", Front. Phys. 8, 127 (2020), would be worthwhile. Apart from this, I can only reiterate my overall very positive impressions and congratulate the authors to a fine contribution.

Reviewer #2: This manuscript proposes a model which considers the aspects (i) the number of contacts of individuals and (ii) age-dependent contact pattern. Four different versions of the model are introduced depended on whether considering the heterogeneity of the two aspects. The performance of the models is examined with SIS and SIR epidemic models based on a data of human contact in Italy. I think this proposed model is interesting. However, this work looks only on its half way which demands a more systematical evaluation under a wide parameter space. The main results presented in the work are based on a specific data and the provided conclusions may not show its generality. Thus, I could not recommend its publication on Plos Computational Biology.

Following are some points for the notice of the authors:

1. The most results shown in this manuscript are based on the parameters induced from the data of Italy. A systematic investigation of the model is recommended from which general properties pertain to the model need to be abstracted and the results of empirical networks could serve as a potent examples.

2. Many concepts and legends in the figures are not well defined. For example, the "Frequency" in Fig. 1D; and is the "number of contacts" should be "age"? What is the 'X' in Fig. 2B? What is the "Relative Difference (%)" in Fig. 2C.

3. Some results in the figures seem not sufficient in its reliability. For example, generally, human contacts are symmetric and reciprocal. However, the gray plot in Fig. 1C looks asymmetric.

4. In addition, in Fig. 3B, what is the definition of R0 according to the age and more importantly how is this age-dependent R0 obtained. Are the results in Fig. 3B theoretical or numerical. If numerical, what is the number of realizations in the simulation and what are the sizes of error bars?

5. The results in Fig. 2C seem to be obtained from the contact pattern in Fig. 1C. However, in the manuscript it is not sufficiently clarified. Since this result presented together with A and B, confusion is easily occurred.

6. Why the study of vaccination is only put on SIR model? Since the SIS model is also an object being addressed in this work, the study of the vaccination should also been applied to the SIS model for the completeness of the work.

7. Some arguments are difficult to understand, e.g. "To gauge the effect ..." in line 300.

8. As mentioned, the results of this manuscript are largely based on a specific data. Therefore, the observations from the data may not suggest a general conclusion. Thus, I don't think the arguments under general terms without mentioning specific conditions, for example "This result might seem ..." in line 312, could explain the specific observations on its above, since in other conditions the observations could be essentially different.

9. "Attack rate" looks not a common term used in the complex network epidemiology.

There are also a few other points need to be addressed, while these here are enough for me come to the conclusion that this manuscript does not meet the standard of Plos Computational Biology.

**Have all data underlying the figures and results presented in the manuscript been provided?**

Reviewer #1: Yes

Reviewer #2: None

PLOS authors have the option to publish the peer review history of their article (what does this mean?). If published, this will include your full peer review and any attached files.

Reviewer #1: No

Reviewer #2: No
---

## [Editor Report · Decision Letter 1]

9 Jun 2020

Dear Dr. Aleta,

We are pleased to inform you that your manuscript 'Data-driven contact structures: from homogeneous mixing to multilayer networks' has been provisionally accepted for publication in PLOS Computational Biology.

Best regards,

Benjamin Althouse

Associate Editor

PLOS Computational Biology

Rob De Boer

Deputy Editor

PLOS Computational Biology

---

## [Editor Report · Acceptance letter]

9 Jul 2020

PCOMPBIOL-D-20-00448R1 

Data-driven contact structures: from homogeneous mixing to multilayer networks

Dear Dr Aleta,

I am pleased to inform you that your manuscript has been formally accepted for publication in PLOS Computational Biology. Your manuscript is now with our production department and you will be notified of the publication date in due course.

With kind regards,

Sarah Hammond
